# Extent of Resection in Newly Diagnosed Glioblastoma: Impact of a Specialized Neuro-Oncology Care Center

**DOI:** 10.3390/brainsci8010005

**Published:** 2017-12-25

**Authors:** Amer Haj, Christian Doenitz, Karl-Michael Schebesch, Denise Ehrensberger, Peter Hau, Kurt Putnik, Markus J. Riemenschneider, Christina Wendl, Michael Gerken, Tobias Pukrop, Alexander Brawanski, Martin A. Proescholdt

**Affiliations:** 1Wilhelm Sander Neuro-Oncology Unit, University Medical Center Regensburg, 93053 Regensburg, Germany; amer.haj@klinik.uni-regensburg.de (A.H.); Christian.Doenitz@klinik.uni-regensburg.de (C.D.); karl-michael.schebesch@klinik.uni-regensburg.de (K.-M.S.); Denise-Ehrensberger@gmx.de (D.E.); Peter.Hau@klinik.uni-regensburg.de (P.H.); Kurt.Putnik@klinik.uni-regensburg.de (K.P.); markus.riemenschneider@ukr.de (M.J.R.); Christina.Wendl@klinik.uni-regensburg.de (C.W.); Michael.Gerken@klinik.uni-regensburg.de (M.G.); Tobias.Pukrop@klinik.uni-regensburg.de (T.P.); alexander.brawanski@klinik.uni-regensburg.de (A.B.); 2Department of Neurosurgery, University Medical Center Regensburg, 93053 Regensburg, Germany; 3Department of Neurology, University Medical Center Regensburg, 93053 Regensburg, Germany; 4Department of Radiation Oncology, University Medical Center Regensburg, 93053 Regensburg, Germany; 5Department of Neuropathology, University Medical Center Regensburg, 93053 Regensburg, Germany; 6Department of Neuroradiology, University Medical Center Regensburg, 93053 Regensburg, Germany; 7Tumor Center Regensburg, Institute of Quality Assurance and Health Services Research, University of Regensburg, 93053 Regensburg, Germany; 8Department of Hematology and Oncology, University Medical Center Regensburg, 93053 Regensburg, Germany

**Keywords:** resection, glioblastoma, fluorescence guidance, functional imaging, outcome

## Abstract

Treatment of glioblastoma (GBM) consists of microsurgical resection followed by concomitant radiochemotherapy and adjuvant chemotherapy. The best outcome regarding progression free (PFS) and overall survival (OS) is achieved by maximal resection. The foundation of a specialized neuro-oncology care center (NOC) has enabled the implementation of a large technical portfolio including functional imaging, awake craniotomy, PET scanning, fluorescence-guided resection, and integrated postsurgical therapy. This study analyzed whether the technically improved neurosurgical treatment structure yields a higher rate of complete resection, thus ultimately improving patient outcome. Patients and methods: The study included 149 patients treated surgically for newly diagnosed GBM. The neurological performance score (NPS) and the Karnofsky performance score (KPS) were measured before and after resection. The extent of resection (EOR) was volumetrically quantified. Patients were stratified into two subcohorts: treated before (A) and after (B) the foundation of the Regensburg NOC. The EOR and the PFS and OS were evaluated. Results: Prognostic factors for PFS and OS were age, preoperative KPS, O^6^-methylguanine-DNA-methyltransferase (*MGMT*) promoter methylation status, isocitrate dehydrogenase 1 (*IDH1*) mutation status and EOR. Patients with volumetrically defined complete resection had significantly better PFS (9.4 vs. 7.8 months; *p* = 0.042) and OS (18.4 vs. 14.5 months; *p* = 0.005) than patients with incomplete resection. The frequency of transient or permanent postoperative neurological deficits was not higher after complete resection in both subcohorts. The frequency of complete resection was significantly higher in subcohort B than in subcohort A (68.2% vs. 34.8%; *p* = 0.007). Accordingly, subcohort B showed significantly longer PFS (8.6 vs. 7.5 months; *p* = 0.010) and OS (18.7 vs. 12.4 months; *p* = 0.001). Multivariate Cox regression analysis showed complete resection, age, preoperative KPS, and *MGMT* promoter status as independent prognostic factors for PFS and OS. Our data show a higher frequency of complete resection in patients with GBM after the establishment of a series of technical developments that resulted in significantly better PFS and OS without increasing surgery-related morbidity.

## 1. Introduction

Glioblastoma (GBM) is not only the most frequent primary brain tumor in adults [1] but it also has an exceptionally poor prognosis [2]. Despite optimal treatment consisting of microsurgical resection followed by concomitant radiochemotherapy and adjuvant chemotherapy, virtually all tumors recur with a median time to progression of 6.9 months, resulting in an average overall survival rate of about 16 months [3]. Although no class I evidence is available, many studies have reported a positive correlation between the extent of resection (EOR) and overall survival in patients with GBM [4]. Some reports using volumetric quantification of EOR have indicated a threshold beyond which a significant survival benefit has to be achieved [5,6]. Yet, the German Glioma Network Study has concluded that only complete resection of the contrast-enhancing lesion is superior to biopsy only [7]. Because of the infiltrative growth pattern, an additional clinical benefit may be obtained by supramarginal resection [8,9,10]. However, the best validated treatment standard is still maximal resection of the contrast-enhancing area. An important limitation to maximal surgical resection is the potential hazard of inducing surgery-related neurological impairment with detrimental effects on patient outcome [11,12]. Taken together, the current neurosurgical paradigm in the management of GBM is maximal resection while avoiding additional neurological morbidity. To achieve this goal, specialized neuro-oncological centers (NOC) have been established that use a large portfolio of diagnostic and therapeutic tools. Optimal resection results are achieved by the implementation of these technical innovations in neurosurgery that should improve patient outcome [13]. The aim of our study was to investigate whether the foundation of a large volume NOC with an optimized technical framework leads to better resection results and improved progression-free and overall survival in patients with newly diagnosed GBM.

## 2. Materials and Methods

### 2.1. Patient Population

The study was approved by the Ethics Committee of the University of Regensburg (protocol 15-101-0065) and conducted in accordance with the ethical guidelines of the Helsinki Declaration. Each patient signed an informed consent form for participation of this trial. Three hundred and ninety-three patients treated for newly diagnosed GBM at the University Medical Center Regensburg were screened between 2005 and 2013. From the screened population, 92 (23.5%) patients received a biopsy only and were excluded. From the remaining 301 patients, preoperative high-quality magnetic resonance imaging (MRI) data for volumetric EOR analysis were available for 149 patients. This cohort was selected for volumetric analysis of pre- and post-operative tumor volume and the resulting EOR. The baseline characteristics of the entire study population are described in Table 1. To address the impact of the NOC foundation, patients were stratified into two groups: Subcohort A was treated before the NOC foundation (June 2005 to June 2009); subcohort B was treated after the NOC foundation (July 2009 to December 2013). During NOC development, neurosurgical strategy was augmented by a series of technical improvements. This consisted of functional MRI, diffusion tensor imaging (DTI)-based fiber tracking, fluorescence-guided resection and awake craniotomy. In addition, postsurgical therapy was optimized by the implementation of a coherent, multidisciplinary treatment matrix consisting of all medical specialties involved in the management of GBM patients [14].

To validate the technical improvement due to the NOC foundation, the frequency of functional imaging, awake craniotomy, fluorescence-guided surgery, and preoperative positron emission tomography (PET) scanning were recorded for both subcohorts. Functional independence and neurological performance were quantified by means of the Karnofsky Performance Score (KPS) and the Medical Research Council Neurological Performance Score (MRC-NPS). Surgically induced morbidity was assessed by worsening of the MRC-NPS score. Follow-up consisting of a review of outpatient records, death certificates from federal registration offices, and contacting the patient’s family or the patient´s primary physician was completed by March 2017. No patient was lost to follow-up, and the median follow-up time was 18.3 months.

### 2.2. Imaging Protocol and EOR Volumetry

MRI imaging was conducted with a 1.5 or 3 Tesla scanner. Imaging sequences consisted of 3D-MPRage iso-volumetric data with 1.0 mm voxel size with and without a gadolinium contrast agent. Postoperative MRIs were obtained within 72 h after surgery. The imaging data were imported into Brainlab iPlan cranial software (version 3.0.2., BrainLAB, Munich, Germany). Subsequently, pre- and postoperative tumor volume was defined based on the gadolinium-enhanced T1 sequences using manual segmentation of all slices. The preoperative tumor volume was assessed by quantifying all tissue with pathological enhancement plus the central necrosis of the tumor. Tumor progression was defined according to revised assessment in neuro-oncology (RANO) criteria as either 25% or more increase of an enhancing lesion; significant increase of a non-enhancing T2/fluid-attenuated inversion recovery (FLAIR) lesion or occurrence of any new lesions. In addition, clinical deterioration not attributable to other non-tumor causes was valued as progression.

### 2.3. Statistical Analysis

Differences in rates and proportions were assessed by means of contingency tables followed by a Pearson’s chi-squared test. Overall survival was analyzed using the Kaplan-Meier method, and Log-rank analysis was used to calculate differences in progression-free and overall survival. Independent predictive factors for survival were isolated by multivariate analysis using Cox hazard regression modelling. Based on the recent literature, we have included the most important prognostic factors into the multivariate analysis: Age, preoperative KPS, *MGMT* methylation status as well as resection status. For multivariate cox regression analysis, we have stratified the parameter KPS into two groups, >/< 70% [15]. Significance was defined as *p* < 0.05 (Stata Version 14.0, Stata Corp., College Station, TX, USA).

## 3. Results

In the entire population, surgical resection had significantly improved both KPS and NPS (*p* = 0.008) compared to the presurgical status. The mean pre-operative tumor volume was 31.4 mL (range: 117.3–1.3 mL), the mean post-operative tumor volume 3.1 mL (range: 5.9–0 mL), resulting in a mean EOR of 90.2% (range: 100–27.4%). 74 patients (49.7%) underwent complete resection, defined by an EOR of 100%. The established prognostic parameters were not significantly different between patients with 100% or <100% resection, except for a trend towards a better preoperative KPS in the 100% EOR group (*p* = 0.073, Table 2). PFS (median 9.4 vs. 7.8 months; *p* = 0.042) and OS (median 18.4 vs. 14.5 months; *p* = 0.005; Figure 1A,B) were significantly better in the 100% EOR group. Univariate analysis revealed age (*p* = 0.006), preoperative KPS (*p* = 0.002), *MGMT* status (*p* = 0.027), 100% EOR (*p* = 0.005) and the treatment period (before or after NOC foundation; *p* = 0.009) as prognostic factors for OS. Multivariate Cox regression analysis established 100% EOR as an independent prognostic parameter for improved OS next to age, treatment period, preoperative KPS, and *MGMT* promoter status (Table 3). We did not observe a significant difference in the KPS and NPS improvement rates between complete and incomplete resection. Most importantly, in the entire cohort the incidence of surgically induced neurological morbidity did not differ between the EOR 100% group and the <100% group (5,4% vs. 4.0%; *p* = 0.983). Comparing the two subcohorts before (A) and after (B), the foundation of a specialized NOC, we observed a profound change in the neurosurgical strategy that consisted of a significant increase in preoperative fMRI scanning (13.1% vs. 41.0%; *p* = 0.002) and fluoroethyl tyrosine (FET)-PET imaging (19.6% vs. 39.8%; *p* = 0.026), and fluorescence-guided resection (8.7% vs. 34.0%; *p* = 0.002). In contrast, the frequency of awake craniotomy did not significantly differ between the two subcohorts (8.7% vs. 6.8%; *p* = 0.944). Interestingly, the change in the neurosurgical treatment pattern was correlated with a significantly increased rate of complete resections in subcohort B (34.8–68.2%; *p* = 0.007, Figure 2) and the mean volumetric EOR (84.3 vs. 96.1%; *p* = 0.01). Finally, although the two subgroups treated before and after NOC foundation were well balanced with regard to the main prognostic parameters (Table 4), PFS (8.6 vs. 7.5 months; *p* = 0.010) and OS (18.7 vs. 12.4 months; *p* = 0.001) were significantly better in subcohort B treated after the NOC foundation (Figure 3A,B).

## 4. Discussion

The treatment of GBM remains a formidable challenge. The extensive efforts made in basic and clinical research have substantially increased the understanding of molecular and pathophysiologic mechanisms but have not significantly improved the clinical management of GBM [16]. New strategies including antiangiogenesis [17] and immunotherapy [18] have not met their initial expectations. Until a therapeutic breakthrough can be achieved, it is of paramount importance to improve the interdisciplinary pattern of care to provide optimal treatment for patients with GBM [13]. Complete microsurgical resection without causing any additional neurological deficits has become the current paradigm, leading to maximal reduction of the malignant cell pool and improving the efficacy of adjuvant treatment efforts [19]. Our data show that the efforts invested in the foundation of a specialized neuro-oncology care center are not only associated with a profound increase in complete resection rates but also with a significant improvement in both PFS and OS. This result is in accordance with several reports on improved overall survival rates in more recently treated patients [20,21,22,23]. However, these studies compared patient cohorts before and after the introduction of a temozolomide-based regimen that may play an important role in the observed improvement in outcome. To investigate the impact of the quality of primary resection within an interdisciplinary course of treatment, we analyzed two subcohorts that were well balanced with regard to the postsurgical treatment pattern.

Our study has a number of limitations. First, stratification according to the NOC foundation is somewhat artificial, since the acquisition of technical improvements was a continuous rather than a stepwise process. Still, we were able to show that the preoperative workup intending to define both the metabolically active tumor area by FET-PET scanning and the optimal resection trajectory by functional imaging had changed significantly after the NOC foundation. In addition, the number of fluorescence-guided resections was significantly increased, improving the quality of resection in GBM [24,25].

Another limitation may be that the relatively small number of patients did not allow for the definition of an EOR threshold for outcome improvement [26]. However, this aspect was not the primary focus of this study. Finally, this study has a retrospective, observational design that may limit the statistical methodology to some degree. In particular, other parameters such as modernized post-surgical treatment regimen may also have an impact on the better outcome after NOC foundation in addition to the improved resection results. However, data were extracted from primary databases of the Tumor Center Regensburg and the University Medical Center Regensburg, so that no patient was lost to follow-up. We are in the process of validating our results by conducting a prospective registry trial as a continuous benchmark process with the intent to improve the interdisciplinary treatment platform for patients with GBM.

## 5. Conclusions

In conclusion, our data show a profound improvement in neurosurgical treatment after the foundation of a specialized neuro-oncology center that resulted in a significant survival benefit in patients with newly diagnosed GBM.

## Figures and Tables

**Figure 1 brainsci-08-00005-f001:**
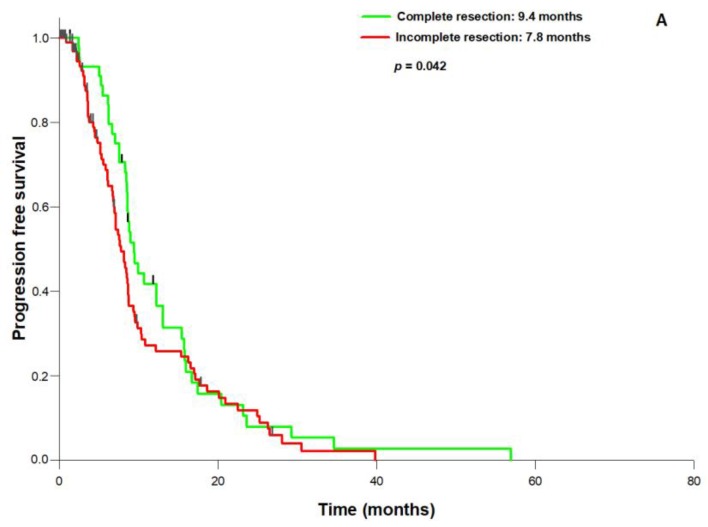
Resection status and outcome in patients with newly diagnosed GBM: (**A**): Progression-free survival is significantly longer after 100% resection than after <100% resection; (**B**): Complete resection also leads to significant overall survival benefit.

**Figure 2 brainsci-08-00005-f002:**
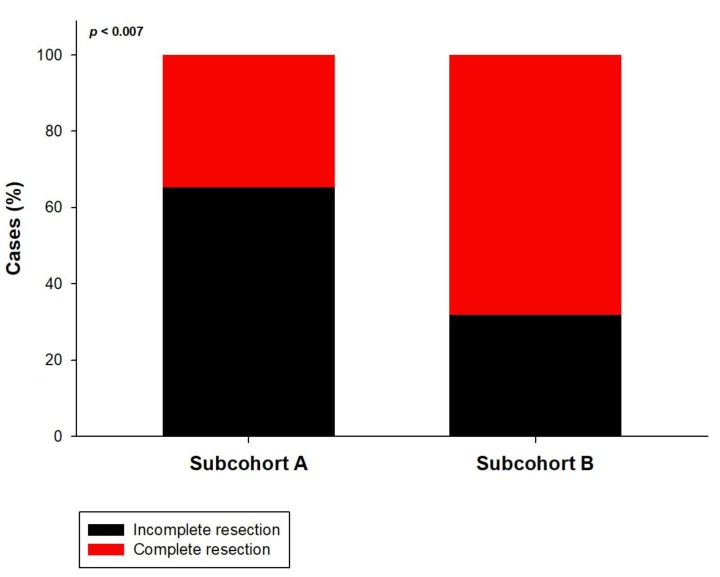
Impact of the NOC (neuro-oncology care center) foundation on the quality of resection: Subcohort (**B**) treated after the NOC foundation showed a significantly higher proportion of complete resections compared to subcohort (**A**) treated before NOC foundation.

**Figure 3 brainsci-08-00005-f003:**
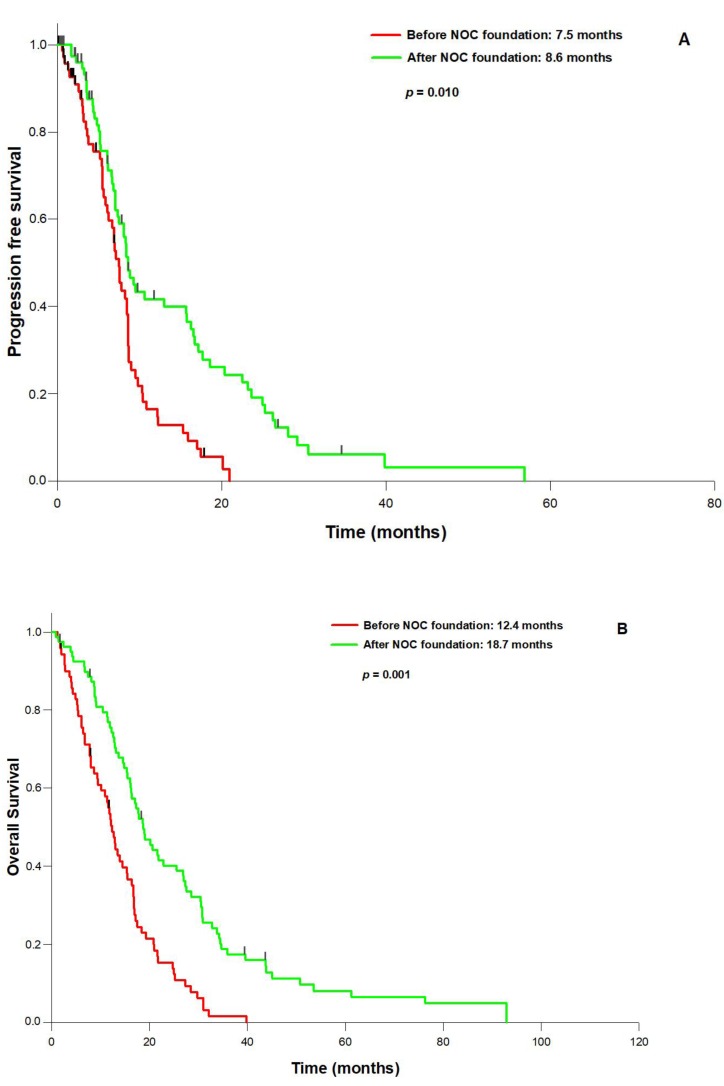
Outcome before and after the NOC foundation: (**A**): progression-free survival and (**B**): overall survival is significantly longer in the subcohort treated after the NOC foundation.

**Table 1 brainsci-08-00005-t001:** Baseline characteristics of the entire study cohort. All patients were surgically treated for newly diagnosed GBM.

Parameter	(Number (%))
*n*	149
Sex (f/m)	67/82 (44.9/55.1)
Age (years, mean)	61.8 (range: 26.7–87.8)
Preoperative KPS (%, median)	80 (range: 20–100)
Preoperative MRC-NPS (points, median)	2 (range: 1–5)
*MGMT* promoter status	Methylated: 50 (33.6)
	Unmethylated: 68 (45.6)
	Unknown: 31 (20.8)
*IDH1* status	Wild type: 144 (96.6)
	Mutated: 5 (3.4)
Postsurgical treatment	
Stupp	106 (71.1)
Radiation only	15 (10.1)
Chemotherapy only	16 (10.7)
No treatment	12 (8.1)

GBM: Glioblastoma; KPS: Karnofsky performance score; MRC-NPS: Medical Research Council Neurological Performance Score; *MGMT*: O^6^-methylguanine-DNA-methyltransferase; *IDH1*: isocitrate dehydrogenase 1.

**Table 2 brainsci-08-00005-t002:** Baseline characteristics of patients with GBM who underwent complete (EOR: 100%) or incomplete resection (EOR < 100%).

Parameter	Complete Resection (Number (%))	Incomplete Resection (Number (%))	*p*
*n*	74 (49.7)	75 (50.3)	
Sex (f/m)	34/40 (46.0/54.0)	33/42 (44.0/56.0)	0.941
Age (years, mean)	61.7 (range: 32.9–80.1)	61.8 (range: 26.7–87.8)	0.939
Preoperative KPS (%, median)	90 (range: 60–100)	80 (range: 20–100)	0.073
Preoperative MRC-NPS (points, median)	2 (range: 1–3)	2 (range: 1–5)	0.219
*MGMT*:			0.746
Methylated	24 (32.4)	26 (34.7)	
Unmethylated	34 (45.9)	34 (45.3)	
Unknown	16 (21.6)	15 (20.0)	
*IDH1* status:			0.988
Wildtype	71 (97.3)	73(97.3)	
Mutated	3 (2.7)	2 (2.7)	
Postsurgical treatment:			0.782
Stupp	55 (74.3)	51 (68.0)	
Radiation only	6 (8.1)	9 (12.0)	
Chemotherapy only	8 (10.8)	8 (10.7)	
No treatment	5 (6.8)	7(9.3)	

EOR: Extent of resection.

**Table 3 brainsci-08-00005-t003:** Multivariate analysis of prognostic factors for overall survival in 149 patients with newly diagnosed GBM.

Parameter	Hazard Ratio	95% CI		*p*
Age	1.032	1.014	1.050	0.001
NOC foundation	0.697	0.482	1.008	0.015
Preoperative KPS (>/<70%)	0.971	0.957	0.988	0.001
*MGMT*	1.47	1.150	1.891	0.002
Resection status: (complete vs. incomplete)	0.981	0.964	0.998	0.032

NOC: neuro-oncology care center.

**Table 4 brainsci-08-00005-t004:** Baseline characteristics of the two patient subcohorts. Group A: Subcohort of patients with newly diagnosed GBM treated before the foundation of a specialized neuro-oncology center (NOC). Group B: Subcohort of patients treated after the NOC foundation.

Parameter	Group A (Number (%))	Group B (Number (%))	*p*
*n*	49 (32.9)	100 (67.1)	
Sex (f/m)	21/28 (42.9/57.1)	46/54 (46.0/54.0)	0.717
Age (years, mean)	62.4 (range: 32.1–81.0)	61.5 (range: 26.7–87.8)	0.647
Preoperative KPS (%, median)	80 (range: 40–100)	80 (range: 20–100)	0.658
Preoperative MRC-NPS (points, median)	2 (range: 1–5)	2 (range: 1–4)	0.68
*MGMT*:			0.431
Methylated	18 (36.7)	32 (32.0)	
Unmethylated	20 (40.9)	48 (48.0)	
Unknown	11 (22.4)	20 (20.0)	
*IDH1* status:			0.889
Wildtype	47 (95.9)	97 (97.0)	
Mutated	2 (4.1)	3 (3.0)	
Postsurgical treatment:			0.195
Stupp	30 (61.2)	76 (76.0)	
Radiation only	8 (16.3)	7 (7.0)	
Chemotherapy only	4 (8.2)	12 (12.0)	
No treatment	5 (10.2)	7 (7.0)	

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
