# Peer review of "Extent of Resection in Newly Diagnosed Glioblastoma: Impact of a Specialized Neuro-Oncology Care Center"

_brainsci, 2017, doi:10.3390/brainsci8010005_

Round 1
Reviewer 1 Report
The authors present a small, retrospective study looking at the impact of the development of a neuro-oncology care centre on the outcomes for patients with GBM. They show that there is an increased rate of complete resections over time and a small but significant improvement in survival. This is an interesting paper that shows a potential impact for patients however some issues need to be addressed:
Firstly the authors describe a multivariate analysis but do not explain which parameters were included in this analysis or why they were included, and not others. They do not give the univariate analysis for any of these parameters. Also, they don't include the formation of the NOC as one of the parameters. Given that this was the one of the main points of the paper this seems to be an oversight.
Secondly, the authors state that there is an improvement in EFS following complete resection, however this improvement is statistical, as the two curves in Figure 1A completely overlap. There is a slight difference at the midpoint, which explains the statistical significance, however the difference is 1 month, and given that the curves then overlap completely this does not seem to be of clinical significance. Moreover, in Figure 3A, the curves have the opposite appearance - that is they initially overlap and then diverge. Therefore, it becomes difficult to conclude that the two are actually related and that the difference in PFS actually relates to resection for those patients treated post formation of the NOC. This discrepancy needs to be addressed.
Third, there is no analysis of post operative performance status following either complete resection or formation of the NOC. They do compare "neurological morbidity" but this is not defined. It would be useful to compare KPS directly. The abstract states that both NPS and KPI improved postsurgically - but this does not seem to be discussed anywhere in the results.
Page 3, line 112 - This sentence (starting "The parameters potentially related to...") does not make sense / is difficult to interpret.
There is nothing detailed about ethics approval, which is critical, especially as the investigators state they have used information from death certificates for follow-up information.
How is PFS defined? What constitutes progression?
Page 4, line 126 - the statement that the the change in neurosurgical treatment "resulted" in an increase in complete resections is an overreach - associated, or correlated would be better terms as retrospectively you cant prove a cause and effect.
English language use needs to be reviewed and improved throughout the manuscript.
Author Response
Firstly the authors describe a multivariate analysis but do not explain which parameters were included in this analysis or why they were included, and not others.
Answer: We thank the reviewer for this suggestion. The most recent studies discussing prognostic factors in newly diagnosed GBM consistently report age, KPS, MGMT promoter methylation status as well as extent of resection (EOR) as parameters significantly associated with overall survival (OS)3, 4, 6, 8. Our rationale to choose these parameters for uni- and multivariate parameters was to stay consistent with the current research in the field. We have added a description of the parameters included in the multivariate analysis in the methods section.
They do not give the univariate analysis for any of these parameters.
Answer: We have added the univariate analysis results in the results section
Also, they don't include the formation of the NOC as one of the parameters. Given that this was the one of the main points of the paper this seems to be an oversight.
Answer: The reviewer is absolutely right, this was an oversight. We have corrected it and added the NOC foundation into the uni- and multivariate analysis (results section, table 3).
Secondly, the authors state that there is an improvement in EFS following complete resection; however this improvement is statistical, as the two curves in Figure 1A completely overlap. There is a slight difference at the midpoint, which explains the statistical significance, however the difference is 1 month, and given that the curves then overlap completely this does not seem to be of clinical significance.
Answer: We agree with the reviewer’s point, that there appears to be a discrepancy between the impact of complete resection on progression free survival (PFS) vs. overall survival (OS). We have two hypotheses regarding this aspect: 1. There is increasing evidence that PFS may not be an entirely valid surrogate marker for OS. In fact, the correlation between the two parameters has been demonstrated to be only moderate5. This is caused a. by a large postprogression heterogeneity1 and b. by the limitations of a imaging – based progression definition in clinical studies as well as in clinical practice especially with regard to pseudoprogression and post – therapeutic alterations of MRI imaging results. 2. In several clinical trials, a complete resection has increased the efficacy of adjuvant, postsurgical treatment7, 9. The major assumption is, that reduction of the malignant cell pool will allow better response with regard to OS but not necessarily to PFS.
Moreover, in Figure 3A, the curves have the opposite appearance - that is they initially overlap and then diverge. Therefore, it becomes difficult to conclude that the two are actually related and that the difference in PFS actually relates to resection for those patients treated post formation of the NOC. This discrepancy needs to be addressed.
Answer: This conclusion follows formal logic: a. Complete resection in GBM patients is associated with longer OS, in our patient population as well as in numerous other studies. b. The implementation of technical improvements has significantly enhanced the proportion of complete resections c. patients treated after the NOC foundation show a significantly longer OS. However, we must agree with the reviewer, that probably other factors, affecting PFS and OS have been improved in addition to the improved resection results. We have added this aspect to the discussion section.
Third, there is no analysis of post operative performance status following either complete resection or formation of the NOC. They do compare "neurological morbidity" but this is not defined. It would be useful to compare KPS directly. The abstract states that both NPS and KPI improved postsurgically - but this does not seem to be discussed anywhere in the results.
Answer: We have stated in the results section (110) that “In the entire population, surgical resection resulted in a significant improvement of both KPS and NPS (p = 0.008) compared to the presurgical status”. We have added an additional analysis, comparing the difference in KPS and NPS improvement rates between complete and incomplete resection, to comply with the reviewer´s request.
Page 3, line 112 - This sentence (starting "The parameters potentially related to...") does not make sense / is difficult to interpret.
Answer: We agree with the reviewer´s point, we have changed the wording of this sentence in accordance to query 1 of reviewer 1.
There is nothing detailed about ethics approval, which is critical, especially as the investigators state they have used information from death certificates for follow-up information.
Answer: This is a very important aspect: We have therefore reported the approval of this study by our local Ethics Committee including the protocol number in the paragraph 2.1. (73-74).
How is PFS defined? What constitutes progression?
Answer: In our study, tumor progression was defined according to RANO criteria as either 25% or more increase of an enhancing leasion; significant increase of a non-enhancing T2/FLAIR lesion or occurrence of any new lesions. In addition, clinical deterioration not attributable to other non-tumor causes was valued as progression. We have included the criteria for progression in the methods section, paragraph 2.2. Imaging protocol
Page 4, line 126 - the statement that the change in neurosurgical treatment "resulted" in an increase in complete resections is an overreach - associated, or correlated would be better terms as retrospectively you cant prove a cause and effect.
Answer: We agree with the reviewer´s suggestion and changed the phrasing of this sentence accordingly.
English language use needs to be reviewed and improved throughout the manuscript.
Answer: We have submitted this manuscript to a linguistic review performed by a professional language editor. We will submit the confirmation of the language editor together with this review response.

Reviewer 2 Report
Review on «Extent of resection in newly diagnosed glioblastoma: impact of a specialized neuro-oncology care center» by Amer Haj et al.
24 : neuro-oncology
24 : Please specify and rephrase : « best » outcome. Concerning which parameter ?
26/27 : integrated postsurgical therapy => please explain in more detail in the methods section
33/34 : Both NPS and KPI significantly improved postsurgically => remove from abstract
37/38 : frequency of transient and permanent postoperative neurological deficits was not higher after complete resection => in both cohorts ? Please mention in results section
44/45 : series of technical developments in the neurosurgical management => please explain in more detail in the methods section ! Was nTMS used ? Was Fiber Tracking regularly done ? Where patients evaluated for neurocognitive or neuropsychological deficits ? Was MEG used ?
65 : neuro-oncology (stay consistent)
76 : why only 149 out of 393 ? You imply that this was do to the lack of high qualitiy imaging. Was biopsy an exclusion criteria as well ?
79 : baseline characteristics are described
Table 1 : add units (years for age, % for KPI), KPI instead of KPS (stay consistent), why is NPS not mentioned ?
93/94 : 1.5 or 3 Tesla scanner is mentioned twice
95 : voxel size
116 : does the group size allow a comparison between 100%EOR vs. Subtotal vs. Incomplete Resection ?
119 : How was surgical morbidity measured ? KPI ? BMRC Grading scale ?
119 : Surgical morbidity was 6% for CRET and 2.7% for subtotal, hence twice as high. Please check p value, and spedify the type of morbidity.
125 : change instead of improvement
130 : were instead of was
Table 2 : add units (years for age, % for KPI), KPI instead of KPS (stay consistent), why is NPS pre- and postop not mentioned ?
Table 3: age as continuous scale variable and KPS as ordinal scale variable can not form a «Hazard ratio». If dichotomization was applied, please explain !
Table 4 : In period B twice as many patients were operated compared to period A, despite the same time frame. Please explain why ? Was this due to more referals or due to paradigm shift in treatment ? Reformat the line IDH Status. When was the Stupp protocol implied at your center (initial publication was in march 2005, patients are included from June 2005)? Why were there only 61.2% of patient treated with Stupp in Group A?
161: is the increase in survival in Group B purely due to an increase in the rate of CRET and increased use of the Stupp protocol or is there a persistent survival benefit after adjusting for those parameters ? Please comment on the relationship between the case load of a center and a surgeon and the outcome
174 : please citate Schucht et al. Neurosurg Focus 2014
General remarks :
- how many of the surgeons changed from period A to period B ?
- how many patients did undergo biopsy only in period A and B ? Biopsies are not mentioned throughout the paper, and one wonders where they were excluded. Patient selection has a significant impact on PFS and OS. For instance, one might argue that implementation of a tumorboard and thus multidisciplinary selection of surgical candidates leads to increased rate of biopsies. Please provide information of the rate of biopsies. Survival rates need to be reported for entire cohorts of patients. Otherwise the authors need to specify that they report exclusively on patients selected for surgery (while detailing the rate of biopsies). Sparing moribund patients the burden of surgery through multidisciplinary patient selection can also be seen as an improvement.
- how often and in which form was intraoperative electrophysiological monitoring and mapping used in period A and B?
- Some patients of cohort B had OS of >10 years (Fig 3B, even though period B started in 2009, 8 years ago. Please explain.
- They authors are congratulated for the increase in the CRET rate from 34.8% to 68.2%, and as expected this is indeed associated with an increase in OS (Fig 1). Fig 3, comparison of periods A and B, shows an even greater increase not only in OS but also in PFS. It appears that surgery is not alone responsible for the increase in survival. Can you elaborate on other changes that have occurred through creation of NOC? After all, many previous studies have shown that improved overall care and better monitoring of patients during follow-up significantly increases survival.
Author Response
Reviewer 2:
24 : neuro-oncology
Answer: We have corrected this spelling error
24 : Please specify and rephrase : « best » outcome. Concerning which parameter ?
Answer: We have re – worded this sentence, and specified the term “outcome” to “outcome regarding progression free and overall survival”
26/27 : integrated postsurgical therapy => please explain in more detail in the methods section
Answer: We have added a detailed explanation of the term “integrated postsurgical therapy”” in the method section including an appropriate reference.
33/34 : Both NPS and KPI significantly improved postsurgically => remove from abstract
Answer: We have removed this sentence from the abstract to comply with the reviewer´s request.
37/38 : frequency of transient and permanent postoperative neurological deficits was not higher after complete resection => in both cohorts ? Please mention in results section
Answer: We have included this important information in both the abstract as well as in the result section
44/45 : series of technical developments in the neurosurgical management => please explain in more detail in the methods section ! Was nTMS used ? Was Fiber Tracking regularly done ? Where patients evaluated for neurocognitive or neuropsychological deficits ? Was MEG used ?
Answer: We have included a more detailed description of the technical improvements which were implemented during the neuro-oncology center development in the method section. In fact, fiber tracking was done routinely in all patients, functional MRI only in a subset of patients, however in a much higher frequency in the subcohort treated after the NOC foundation (13.1% vs. 41.0%).Neuropsychological testing was part of the pre- and postoperative work up in patients with either neurocognitive deficits or lesions located in eloquent brain areas. In contrast, although we have used nTMS for a short period of time, this technique was not integrated as part of the routine management.
65 : neuro-oncology (stay consistent)
Answer: We have corrected this spelling error to stay consistent throughout the text.
76 : why only 149 out of 393 ? You imply that this was do to the lack of high qualitiy imaging. Was biopsy an exclusion criteria as well ?
Answer: Since volumetric assessment of resection quantity was one of the major read – outs, we agree with the reviewer comment that biopsy only patients were excluded. That significantly contributes to the reduction of patients we could enroll into the study. We have included this algorithm into the method / patient population section.
79 : baseline characteristics are described
Answer: We have corrected this spelling error
Table 1 : add units (years for age, % for KPI), KPI instead of KPS (stay consistent), why is NPS not mentioned ?
Answer: We have added the units in all tables for age and KPI. In addition, we have included the MRC – NPS scores in the entire population as well as in the subcohorts (EOR 100% and less; subcohort A and B). The subcohorts are balanced for this parameter as well. We thank the reviewer for this important suggestion.
93/94 : 1.5 or 3 Tesla scanner is mentioned twice
Answer: We have corrected this error.
95 : voxel size
Answer: We have corected this error
116 : does the group size allow a comparison between 100%EOR vs. Subtotal vs. Incomplete Resection ?
Answer: Although this is a very important aspect, the group size did not allow a statistical comparison of three groups, stratified into complete, subtotal and incomplete resection.
119 : How was surgical morbidity measured ? KPI ? BMRC Grading scale ?
Answer: The surgical morbidity was measured by the worsening of the BMRC grading scale. We have included this information in the method / patient population section.
119 : Surgical morbidity was 6% for CRET and 2.7% for subtotal, hence twice as high. Please check p value, and spedify the type of morbidity.
Answer: We thank the reviewer for this suggestion. There was a notation error. In the complete resection group 4 patients out of 74 (5.4%) showed morbidity, whereas in the incomplete resection group 3 patients out of 75 (4.0%) showed morbidity. The p – value of the rates and proportions analysis is actually correct. We have corrected this mistake.
125 : change instead of improvement
Answer: We have changed the wording in accordance to the reviewer´s suggestion
130 : were instead of was
Answer: We have corrected this error accordingly.
Table 2 : add units (years for age, % for KPI), KPI instead of KPS (stay consistent), why is NPS pre- and postop not mentioned ?
Answer: We have corrected KPI instead of KPS and checked the entire text for consistency regarding this abbreviation. Also, we have included the MRC – NPS score into the table.
Table 3: age as continuous scale variable and KPS as ordinal scale variable can not form a «Hazard ratio». If dichotomization was applied, please explain !
Answer: We have in fact to stay consistent with the multivariate analysis approach dichotomized the KPS according to better or worse than 70%. We have included this important aspect into table 3.
Table 4 : In period B twice as many patients were operated compared to period A, despite the same time frame. Please explain why ? Was this due to more referals or due to paradigm shift in treatment ?
Answer: There are two explanations for this effect: 1. There were less patients with high quality preoperative imaging available for volumetric analysis in group A. 2. Due to the clinical focus on neuro-oncology during the foundation of this center, we have received significantly more patient referrals in the later period.
Reformat the line IDH Status.
Answer: We have reformatted the line IDH1 status this table
When was the Stupp protocol implied at your center (initial publication was in march 2005, patients are included from June 2005)? Why were there only 61.2% of patient treated with Stupp in Group A?
Answer: That is a very interesting question, yet hard to answer. The implementation of the Stupp protocol in our center started in July 2005. However, this was a continuous process, with increasing proportions of patients receiving Stupp protocol treatment over time. This may explain why only 61.2% of all patients were treated accordingly. In addition, although there was no statistically significant difference in age, preoperative MRC-score and KPS, it is possible, that during that initial time period, elderly patients with comparably poor overall status may not have received treatment according to the Stupp protocol. This is indicated by the fact that a higher percentage (however not statistically significant) in group A received radiation only compared to group B.
161: is the increase in survival in Group B purely due to an increase in the rate of CRET and increased use of the Stupp protocol or is there a persistent survival benefit after adjusting for those parameters ?
Answer: Although it is possible that the higher percentage of patients treated according to the Stupp protocol in group B may have influenced the better survival outcome, there was no statistically significant difference between the two groups with regard to postsurgical treatment. We have attempted to include a formal interaction analysis for type of treatment in the regression modelling, which was also not statistically significant. However, for an adequate interaction analysis we did not have sufficient statistical power with our limited number of patients.
Please comment on the relationship between the case load of a center and a surgeon and the outcome.
Answer: We absolutely agree with the reviewer´s opinion, that there is a clear relationship between case numbers and treatment outcome in glioblastoma. This was already demonstrated in a previous study, analyzing the effects of a high volume center treatment on patients’ outcome2. Although the team of surgeons remained almost completely stable in our department, the higher case number might induce a significant learning curve improving the surgical procedure, avoiding complications and reducing surgically induced morbidity.
174 : please citate Schucht et al. Neurosurg Focus 2014
Answer: We have included this reference in the discussion section
General remarks :
- how many of the surgeons changed from period A to period B ?
Answer: As already mentioned in the previous query, we have had only very moderate changes in our team during the observation period (from 8 attending neurosurgeons, only one was replaced).
- how many patients did undergo biopsy only in period A and B ? Biopsies are not mentioned throughout the paper, and one wonders where they were excluded. Patient selection has a significant impact on PFS and OS. For instance, one might argue that implementation of a tumorboard and thus multidisciplinary selection of surgical candidates leads to increased rate of biopsies. Please provide information of the rate of biopsies. Survival rates need to be reported for entire cohorts of patients. Otherwise the authors need to specify that they report exclusively on patients selected for surgery (while detailing the rate of biopsies). Sparing moribund patients the burden of surgery through multidisciplinary patient selection can also be seen as an improvement.
Answer: We have excluded patients undergoing biopsies since we wanted to quantify the extent of resection. We have included this algorithm in the methods/patient population section and reported the fact, that this study exclusively reports outcome in patients who received resection, not biopsy. We absolutely agree with the reviewer´s opinion, that patient selection for resection vs. biopsy plays an important role. We have checked the rate of biopsies in the two subcohorts, expecting that the rate of biopsies would be reduced in group B. This perception, however was not correct, the rate of biopsies in subcohort A was 26.5% and in subcohort B 21.9%, which was not significantly different (p = 0.537).
- how often and in which form was intraoperative electrophysiological monitoring and mapping used in period A and B?
Answer: Unfortunately, the records regarding electrophysiological monitoring were rather incomplete, not allowing any kind of adequate analysis. All patients undergoing awake craniotomy, however, received neurophysiological monitoring These numbers, did not differ significantly between group A and group B (8.7% vs. 6.8%; p=0.944).
- Some patients of cohort B had OS of >10 years (Fig 3B, even though period B started in 2009, 8 years ago. Please explain.
Answer: This was a notation error regarding two patients. We corrected this error in figure 3 B. The log analysis results and the median OS results were not affected by this error.
- They authors are congratulated for the increase in the CRET rate from 34.8% to 68.2%, and as expected this is indeed associated with an increase in OS (Fig 1). Fig 3, comparison of periods A and B, shows an even greater increase not only in OS but also in PFS. It appears that surgery is not alone responsible for the increase in survival. Can you elaborate on other changes that have occurred through creation of NOC? After all, many previous studies have shown that improved overall care and better monitoring of patients during follow-up significantly increases survival.
Answer: The reviewer has a very valid point. Although the initial postsurgical treatment regimen between group A and B are balanced, changes in the post-progression treatment may also have an impact on the OS benefit. We have therefore checked this aspect and found a significantly higher proportion of second – line bevacizumab treated patients in group B (p = 0.044), a higher frequency of re-resections (0.038) and re – irradiations (p = 0.025). However, since the PFS is significantly higher, which reflects the impact of the initial treatment quality including the extent of resection, we would conclude that the higher percentage of complete resections in the modern era at least significantly contributes to the observed PFS and OS benefit in group B.
